# Job Satisfaction Level of Safety and Health Manager in Construction Industry: Pandemic Period

**DOI:** 10.3390/ijerph19105858

**Published:** 2022-05-11

**Authors:** Won Choi, Sang-joon Lee, Woo-je Lee, Eun-mi Beak, Ki-youn Kim

**Affiliations:** 1Graduate School of Safety Engineering, Seoul National University of Science and Technology, Seoul 01811, Korea; wwclo@naver.com (W.C.); lsj9897@samyoungco.com (S.-j.L.); 34uj@naver.com (W.-j.L.); 2Department of Preventive Medicine, College of Medicine, Catholic University of Korea, Seoul 06591, Korea; hanel2004@naver.com; 3Department of Safety Engineering, Seoul National University of Science and Technology, Seoul 01811, Korea

**Keywords:** COVID-19, safety manager, health manager, job satisfaction, job stress

## Abstract

In a widespread social turmoil such as the Pandemic, job groups with high turnover rates and high job stress, such as the construction industry, will have a greater adverse impact than the general job group. This is to be used as basic data in preparing management plans by identifying the factors that hinder job stress and job satisfaction of construction workers. In this study, during the Pandemic period (1 September 2021 to 31 December 2021), a survey was conducted on job stress and job satisfaction among safety and health managers working at construction sites. The overall job satisfaction of workers in the construction industry was grasped by analyzing the level of correlation and the mutual influence on job stress, job satisfaction, general characteristics, and work-related characteristics. As a result, in terms of work characteristics, it was found that the smaller the working period in the current position, the more positive the job satisfaction was (*p* < 0.01). In addition, it was found that job satisfaction increased significantly when there was a promotion opportunity (*p* < 0.001). The construction industry is a job group with high basic job stress and low job satisfaction. In addition, it was evaluated that job stress increased during the pandemic.

## 1. Introduction

The pandemic period caused by the COVID-19 virus epidemic is the largest global crisis in decades and has caused widespread socioeconomic confusion. On 11 March 2020, the WHO declared it a “Pandemic,” which means a global pandemic. Currently, deaths and hospitalizations related to the unprecedented COVID-19 virus transmission are increasing worldwide. Beyond health threats, the pandemic has caused economic stagnation, widespread business suspension, and severe economic activity difficulties [1]. The United States declared that it entered an economic recession in February 2020. In the United States, it was named “Recession of COVID-19” [2]. The U.S. recorded an all-time high unemployment rate of 14.7% in April 2020 due to the economic recession. This is very low compared to the unemployment rate of 3.8% in February 2020 [3]. This high unemployment rate stems from the devastation of all industries, including aviation, restaurants, manufacturing, and retail, and the reduction of business size.

In addition, the pandemic has changed the organizational structure of the company. In addition to reducing employees’ working hours, work activities should be significantly reduced [4]. This situation has generally caused a decline in productivity and organizational competitiveness [5]. Along with the decline in the competitiveness of the organization, employment anxiety is natural, and job stress also occurs in a series [6]. Pandemics cause instability in employment, resulting in a decrease in job satisfaction [7,8]. If this pandemic period is prolonged, workers’ anxiety and anxiety will not be able to relax, reducing their confidence in their jobs [9].

Like other industries, the construction industry was greatly influenced in many ways. We found construction project suspension, manpower shortage, construction time and cost excess, construction material supply shortage due to factory closure, construction site planning and schedule suspension, and movement restrictions due to the declaration of the pandemic period [10]. According to the construction industry employment data reported by the U.S. Bureau of Labor Statistics (BLS), many construction workers have tested positive for COVID-19. In fact, a recent survey in Los Angeles reported that construction workers had more positive cases than workers employed in other industries such as medical, transportation, and manufacturing [11]. In addition, it was found that construction workers are about five times more likely to be hospitalized due to COVID-19 than workers in other industries [12]. Public ministries in various states in the United States have also stressed that there is a high risk of COVID-19 infection, especially among construction workers [13]. In addition, infectious diseases damaged the income and mental health of construction workers, interfered with project schedules, and incurred costs [14].

The construction industry has contributed greatly to the development of the country from industrialization to the present [15]. Nevertheless, the construction industry continues to suffer from high turnover rates [16]. In order to reduce this turnover rate, the easiest and fastest applicable solution is to identify and adjust the factors contributing to the turnover rate [17]. 

Likewise, in the construction industry, workers’ mental health in the workplace is a very important factor in organizational management [18]. Positive satisfaction with one’s job enables the improvement of the work atmosphere and has a positive effect on work efficiency [19]. In particular, changes in risk perception such as workers’ anxiety, stress, solitude, anxiety, stigma, and discrimination due to the pandemic increased significantly [18,19,20,21,22,23].

In a widespread social turmoil such as a Pandemic, previous studies have reported that occupational groups with high turnover rates and high job stress, such as the construction industry, have a greater adverse effect than general occupational groups [24]. It is necessary to study the impact of the pandemic period on the construction industry. However, previous studies mainly focused on collecting information on the impact of the COVID-19 pandemic from the perspective of construction personnel and its impact on the construction sector and economy [8].

This study investigated job stress and job satisfaction of construction field workers currently undergoing a pandemic period through a questionnaire survey. As there would be many gaps between field and office workers in the construction industry, the safety and health managers who were conducting the field and office work together were selected as the subjects of the survey. In the Korean construction industry, almost all office workers are Korean, and many outdoor workers are made up of foreign workers. In Korea, safety managers and health managers are separated, and their duties are very different. Therefore, the research was conducted based on the judgment of health managers and safety managers for jobs that are easy to collect survey data and can replace both office and outdoor workers. Therefore, this study targeted the construction industry, which had very little prior research on strong work intensity, low job satisfaction, and work environment compared to other industries. The study’s target occupation group included health managers and safety managers who were responsible for both indoor and outdoor activities and information on the overall working environment of the construction site and job satisfaction of all workers. The purpose of this study is to determine the factors affecting job satisfaction during the disaster period by evaluating the job satisfaction of construction health and safety managers in special situations such as a pandemic, and to infer the overall job satisfaction and major factors based on the results.

## 2. Method

### 2.1. Subject

The questionnaire survey of safety and health managers registered with the Korean Academy of Construction and Health was done from September 2021 to December 2021. This study was approved in advance after deliberation by the IRB (Institutional Review Board) of Seoul National University of Science and Technology (2021-0021-01). For the recruitment of research subjects, with the help of the Korean Construction and Health Association, 1454 participants were recruited by e-mail. Only members who wanted to participate in the survey were allowed to participate in the preparation. When distributing the questionnaire by e-mail, the contents of this study, data usage, and disposal methods were sufficiently explained. In order to ensure anonymity once again, the prepared anonymous questionnaire was collected by the conference and delivered to the research manager. The consent of the study subjects was considered as consent to recruit voluntary applicants, read the contents of the study sent by mail, use and discard data at the top of the questionnaire, check the consent column, fill out the survey, and submit it by mail and email. As for the criteria for dropping out of the study, if the survey question was less than 80%, the dropout was processed. The number of subjects required for the study was calculated using the G*Power 3.1, Erdfelder, Faul, & Buchner, Germany [25]. A total of 200 surveys were required when ANOVA statistical analysis, effect size f = 0.25, aerr prob = 0.05, and Power = 0.8 were applied. In consideration of the dropout rate, the questionnaire was distributed to 300 people and the final 227 survey data were collected.

### 2.2. Survey Item

The questionnaire used in this study consisted of items related to general characteristics, work-related characteristics, job stress, and job satisfaction (refer to Appendix A).

#### 2.2.1. General and Work-Related Characteristics

General characteristics included gender, age, marital status, and education level. The work-related characteristics consisted of positions, positions, construction period, current position working period, department, employment type, average weekly working hours, promotion opportunities, during work accidents, and construction site locations.

#### 2.2.2. Job Stress

For job satisfaction, a job satisfaction survey by [26], was used by modifying the translation of ‘The index of work Satisfaction’ developed by [27]. Each question was composed of a 5-point scale that it was not at all, it was not, it was normal, it was, and it was very yes, and the higher the score, the higher the job satisfaction. The reliability Cronbach’s alpha value of the job satisfaction survey is 0.898.

#### 2.2.3. Job Satisfaction 

For job satisfaction, the job satisfaction questionnaire of [26], which was modified and used by ‘The index of work Satisfaction’ developed by [27], was used. Each question was composed of a 5-point scale that it was not at all, it was normal, it was yes, and it was very yes, and the higher the score, the higher the job satisfaction. The reliability Cronbach’s alpha value of the job satisfaction survey is 0.898.

### 2.3. Data Analysis

Frequency analysis was conducted on the general characteristics and work-related characteristics of the survey participants. The level of job stress and job satisfaction related to general characteristics and work-related characteristics were independent sample *t*-test and one-way distribution analysis. Hierarchical multiple regression analysis was conducted to evaluate the general characteristics and work-related characteristics that affect job stress and job satisfaction. As for the significance level, *p* < 0.05 was a significant factor, ** *p* < 0.01 was a significant and strong factor, and *** *p* < 0.001 was a significant and very strong factor. All statistical analyses applied IBM’s SPSS 25.0 version program.

## 3. Results

### 3.1. General and Work-Related Characteristics

Table 1 presents the general and work-related characteristics of the study subjects. In terms of the general characteristics of the subjects, men accounted for 74.0%, and those aged 30–38 accounted for the most at 31.3%. As for marriage, 53.3% of the subjects of the study were unmarried, and college graduates accounted for the highest education level at 74.9%. As for work-related characteristics, 67.8% were safety managers and 31.8% were health managers. The positions were 16.3% for managers, 43.2% for middle managers, and 40.5% for working-level managers. In the construction industry, 44.9% of them worked less than 5 years, and 65.6% of them worked less than 5 years in their current positions. Most of the departments were safety and health departments with 99.6%. As for the employment type, 62.6% of contract workers and 36.1% of regular workers. The average weekly working hours were the highest at 63.9% of 49–56 h per week. 44.1% of them had promotion opportunities, but 41.8% had limitations, and 87.7% of accidents while working, most of them had no accident experience. The location of the construction site was 40.6% in Gyeonggi-do, 16.3% in Seoul, 8.4% in Busan, and 7.1% in Incheon (refer to Table 2).

### 3.2. Job Stress and Job Satisfaction According to General Characteristics

As indicated in Table 2, among the safety and health managers working in the construction industry, both men and women showed similar job stress and job satisfaction, but were not statistically significant. In the age variable, job stress was the lowest among those under the age of 30, and job stress from 30 to 50 years of age was all similar, and statistically significant. Job satisfaction was the highest score in those under the age of 30, and the lowest satisfaction in those aged 30 to 39, but was not statistically significant. In terms of education level variables, the lowest job stress and highest job satisfaction were shown among high school graduates, but they were not statistically significant. In the marital status variable, the highest job stress and lowest job satisfaction were shown when married than when unmarried, divorced, bereavement, or separated, and statistically significant (refer to Table 2).

### 3.3. Job Stress and Job Satisfaction According to Work-Related Characteristics

As shown in Table 3, a total of 227 people responded to the survey, including 72 health managers, 154 safety managers, and one other person. The average for job stress evaluation scores was 57.14 for health managers and 55.50 for safety managers, indicating that health managers have relatively higher job stress than safety managers. On the other hand, the average for job satisfaction was 63.18 for health managers and 65.29 for safety managers, indicating that safety managers were relatively higher than safety managers. In other words, it was found that health managers were less satisfied with their jobs than safety managers, but they were not statistically significant (*p* > 0.05). When the position was a manager, job stress was higher than that of middle managers and working-level officials, and it was statistically significant (*p* < 0.05). On the other hand, in terms of job satisfaction, middle managers were the highest and managers were the lowest, but they were not statistically significant (*p* > 0.05). When the construction industry worked for more than 20 years, job stress was the highest, when it was less than 5 years, and statistically significant (*p* < 0.05). Job satisfaction was the highest when the working period was less than 5 years, and the lowest when it was more than 20 years, but was not statistically significant (*p* > 0.05). Like the construction industry’s working period, the job stress was the highest when it was more than 20 years, the job satisfaction was the highest when it was less than 5 years, and statistically significant (*p* < 0.05). In terms of employment type, contract workers were about 1.7 times more than regular workers, and contract workers had lower job stress and higher work satisfaction than regular workers, and were statistically significant (*p* < 0.05). The more working hours, the higher the working stress, so it was the highest when working more than 57 h. Work satisfaction was high when it was 41 to 48 h and 57 h or more, but it was not statistically significant (*p* > 0.05). Safety and health managers in the construction industry showed the lowest job stress when they had a promotion opportunity, and when they had a promotion opportunity but had a limit, or when there was no promotion opportunity, they showed high job stress and were statistically significant (*p* < 0.05). Job satisfaction was also high when there was a promotion opportunity. It was found that job satisfaction decreased when there was a promotion opportunity but there was a limit and when there was no promotion opportunity, and it was statistically significant (*p* < 0.05). It was about eight times more than the group with no accident experience while working, less job stress, and higher job satisfaction, but was not statistically significant (*p* > 0.05). Most of the construction sites were located in Seoul, and both job stress and job satisfaction were moderate compared to other regions, but were not statistically significant (*p* > 0.05) (refer to Table 3). 

### 3.4. Effect of Job Satisfaction on Job Stress after Controlling Demographic Characteristics

After controlling exogenous variables, hierarchical regression analysis was conducted using general characteristics and work-related characteristics as control variables to find out whether job satisfaction affects job stress. (Model 1) identified the effect of general characteristics and work-related characteristics as control variables on job stress, and (Model 2) added independent variable job satisfaction to find out if job satisfaction affects job stress even after exogenous variable control. As a result of the analysis, it can be said that the regression model is suitable. F = (Model 1) F = 3.065 (*p* < 0.001), (Model 2) F = 11.557 (*p* < 0.001). The R-square change from (Model 1) = 0.294 to (Model 2) = 0.620 increased by 0.326. The significance probability *p* = 0.000 according to the R-square F change amount (F = 1700.389) can be said to be statistically significant in explaining the dependent variable after the control variable is input. 

In (Model 1), statistically significant variables had a negative effect on job stress when there was a promotion opportunity from promotion opportunity in job-related characteristics. In other words, job stress was significantly lower in the promotion opportunity variable (*p* < 0.05).

In (Model 2), statistically significant variables had a negative effect when unmarried in the marital status variable in general characteristics, and a working person had a negative effect on the position in the work-related characteristics. In other words, job stress was significantly lower when they were unmarried and in charge of practical affairs (*p* < 0.05) (refer to Table 4).

### 3.5. Effect of Job Stress on Job Satisfaction after Controlling Demographic Characteristics

After controlling exogenous variables, hierarchical regression analysis was conducted using general characteristics and work-related characteristics as control variables to find out whether job stress affects job satisfaction. (Model 1) identified the effect of general characteristics and work-related characteristics as control variables on job satisfaction, and (Model 2) identified whether job stress affects job satisfaction even after control of exogenous variables. As a result of the analysis, it can be said that the regression model is suitable. F = (Model 1) F = 3.491 (*p* < 0.001), (Model 2) F = 12.317 (*p*< 0.001).

The R-square change from (Model 1) = 0.321 to (Model 2) = 0.635 increased by 0.314 The significance probability *p* = 0.000 according to the R-square F change amount (F = 1700.389) can be said to be statistically significant in explaining the dependent variable after the control variable is input. 

In (Model 1), statistically significant variables were found to have low work satisfaction when they were under 39 years of age among general characteristics (*p* < 0.05). And regardless of marital status, job satisfaction was found to have a negative effect (*p* < 0.01). In terms of work characteristics, it was found that work satisfaction was high within 10 years in the current position (*p* < 0.01). In addition, even if there is a limit to the promotion opportunity, it was found that work satisfaction had a positive effect when there was a promotion opportunity (*p* < 0.05). 

In (Model 2), the statistically significant variable had a statistically significant negative effect on work satisfaction in the marital status variable among general characteristics (*p* < 0.001). In terms of work characteristics, it was found that the smaller the working period in the variable of the working period in the current position, the more positive the work satisfaction was (*p* < 0.01). In addition, it was found that job satisfaction significantly increased when there was a promotion opportunity (*p* < 0.001) (refer to Table 5).

## 4. Discussion

Job satisfaction evaluation can be a measure that can predict the achievement of organizational performance, turnover rate, absence from work, service period, and organizational goal [28]. Studies on job satisfaction so far have concluded that job-related characteristics can have an important effect on both job stress and job satisfaction [29,30].

To interpret the results of this study, among the general factors affecting job stress and job satisfaction, job stress was low and job satisfaction was high at age under 30. And at the age of 30 to 39, stress increased and job satisfaction tended to decrease. This shows a tendency very similar to the marital status factor. In addition, job stress decreases and job satisfaction tend to increase when divorce/divorce/separation makes it free from the burden of family support again. When interpreting the results. As a result, the most prominent factor in job satisfaction in general characteristics is age and marital status, and unmarried people under the age of 30 were free from the burden of family support, but they got married when they were over 30, and job stress increased. In a study on job satisfaction before the pandemic, there were also research results that marital status and family relationship factors influenced job satisfaction [31]. This result would have been further influenced by the overlapping of the pandemic period, increasing employment anxiety. In order to solve these factors, job satisfaction can be increased by solving the requirements for married people at the company welfare level. For example, parental leave, expanding the child education and welfare system, and expanding the scope of health insurance coverage can be alternatives.

Among the work-related factors affecting job stress and job satisfaction, significant results were found in the factors of position, number of years of work, employment type, and promotion opportunity. The higher the number of years of work, the higher the job stress, the lower the job satisfaction, the lower the position, the lower the number of years of work, and the lower the job stress and job satisfaction when the employment type was a contract job. These results need to consider the characteristics of the construction industry. It is judged that there will be a tendency to avoid responsibility for jobs in jobs that have a very high intensity of work such as the construction industry and have a low attachment to jobs. Therefore, job satisfaction would have decreased when the number of working years increased and the position increased and responsibility was required. As a way to solve this problem, it is judged that the factor of promotion opportunity will be an important solution. When there was an opportunity for promotion, workers in the construction industry showed high job satisfaction. Promotion opportunities are an important factor in determining job satisfaction [32]. And as the number of years of work increases, your position increases, which will naturally give you responsibility within the company. In addition, as the number of years of work increases, if a method of converting contract workers into full-time workers is added, job satisfaction of construction workers is expected to increase.

The limitation of this study is that safety and health managers selected as subjects of the study will inevitably increase job stress because they overlap with quarantine issues during the pandemic. In this study, items on salary were not included in the survey. In Korea, salary in the workplace is very personal information. The Korean Construction and Health Association, which distributed and collected the survey, wanted to delete the salary to protect the privacy of workers, but the salary is one of the most important variables contributing to job satisfaction [33]. It is necessary to interpret the results of job satisfaction in consideration of changes in salaries or payments during the pandemic period. In addition, among the parameters that contribute to job satisfaction and job stress during the pandemic, promotion opportunities were found to be significant variables. This will be an important variable for job satisfaction even in non-pandemic situations or other occupations other than the construction industry [34]. In addition, if additional research is conducted on job satisfaction in the construction industry, it is judged that selecting outdoor and indoor workers separately for the study can obtain statistically significant values. In addition, when looking at the current education level of the construction industry in terms of general characteristics and work-related characteristics, the education level was more than 80% of college graduates. This is judged that dissatisfaction with Korea’s high-level education in poor job environments such as the construction industry will have a negative effect on overall job satisfaction. In terms of work-related characteristics, in manufacturing or general industries, as the working period increases and the position increases, the rights, and work environment within the company improve [23]. In the case of the construction industry, however, the increase in working period and high positions only expand the responsibility of workers while they do often not guarantee a good working environment. As a result, as the working period increases, it would act as the cause of high job stress and low job satisfaction.

In this study, regular workers had a 2.46 higher job stress average than contract workers, while the job satisfaction average was 2.42 lower, indicating that regular workers were more dissatisfied with their jobs than non-regular workers. This needs to be interpreted based on the aspect of “employment stability” [35]. In general, regular workers have higher job satisfaction than contract workers [36]. However, the results of this study, like many previous studies, do not show a consistent relationship between contract type and job satisfaction [37,38]. The reasons for this would be high work intensity, high turnover rate, and poor work environment even though they are regular workers due to the nature of the construction industry.

In general, trust in the organization affects job satisfaction [39]. However, due to the nature of the construction industry, trust in the organization is not high, which negatively affects job satisfaction. In addition, in the case of safety and health managers in the construction industry, there are many long-term business trips. Frequent changes in the working environment due to location distribution due to business trips will cause lower reliability among workers [40]. The high turnover rate due to the nature of the construction industry is also expected to have a negative impact on the formation of relationships in organizations [41]. Further research will improve job satisfaction of construction workers and consequently reduce the turnover rate by preparing measures to increase employee safety, organizational reliability, and promotion opportunities. 

Studies that individually evaluated job stress and job satisfaction of construction safety and health managers have already been reported [42,43,44]. This study was the first study to compare the job stress of construction safety and health managers with the resulting job satisfaction level, and although it was not statistically significant, it was analyzed that health managers had relatively higher job stress than safety managers. Most health managers in the construction industry play their role, and it is judged that job satisfaction is low due to excessive job stress caused by the relatively poor and rough working environment conditions in the construction industry.

This study has a limitation in narrowing the scope of interpretation because it is a cross-sectional study that limits the research area, subject, and time of study due to the nature of all survey studies [45]. Since the research was conducted only in the construction industry, the research should be expanded to other occupations with similar strength and job satisfaction [23]. In addition, since the study was conducted in Korea during the temporary pandemic period, additional research including the contents of other countries is needed to confirm the objectivity of the research results in the future.

## 5. Conclusions

As a result of evaluating the job satisfaction level of safety and health managers working in the construction industry during the pandemic period due to COVID-19 virus infectious diseases, it was the highest among unmarried, divorced, bereavement, and separated. The higher the rank, the higher the job stress, and the higher the job stress when the construction industry worked for a long time. In addition, job satisfaction was high when there was an opportunity for promotion, and in terms of employment type, job satisfaction was not high for both regular and contract workers. Therefore, based on the results of this study, safety and health managers in the construction industry are job groups with high basic job stress and low job satisfaction, regardless of general characteristics or work-related characteristics. In addition, the overall working environment of the construction industry needs to be improved to reduce job stress because it is evaluated to increase during the pandemic period. In addition, in jobs such as the construction industry, which have very strong work intensity and low employment safety, it is necessary to frequently survey job stress and job satisfaction during special periods such as the pandemic period, and to understand the psychological state of workers by detailed surveys.

## Figures and Tables

**Table 1 ijerph-19-05858-t001:** General characteristics and work-related characteristics of the study subjects.

Classification	Category	People	%
Sex	Man	168	74.0
	Woman	59	26.0
Age	Under 30	66	29.1
	30~39	71	31.3
	40~49	69	30.4
	Over 50	21	9.2
Education	high school	4	1.7
	College	39	17.2
	University	170	74.9
	Graduate school	14	6.2
Marriage	Single	121	53.3
	Married	97	42.7
	Divorce/Bereavement/Separate	9	4.0
Qualification	Health manager	72	31.8
	Safety manager	154	67.8
	Etc.	1	0.4
Job title	Manager	37	16.3
	Middle manager	98	43.2
	Worker	92	40.5
Construction career	Less than 5 years	102	44.9
	5~10 years	42	18.5
	10~20 years	59	26.0
	More than 20 years	24	10.6
Present career	Less than 5 years	149	65.6
	5~10 years	42	18.5
	10~20 years	26	11.5
	More than 20 years	10	4.4
Department	Health and safety	226	99.6
	General affairs	0	0
	Nursing	0	0
	Infrastructure	0	0
	Etc.	1	0.4
Employment type	Full-time worker	82	36.1
	Temporary worker	142	62.6
	Dispatch labor	0	0
	Part-time worker	0	0
	Etc.	3	1.3
Average working hours per week	Less than 40 h	6	2.6
	41~48 h	52	22.9
	49~56 h	145	63.9
	More than 57 h	24	10.6
Promotion opportunity	Possible	100	44.1
	Possible, but limited	95	41.8
	Impossible	32	14.1
Accident at work	Yes	28	12.3
	No	199	87.7
Location city	Gyeonggi-do	92	40.6
	Seoul	37	16.3
	Busan	19	8.4
	Incheon	16	7.1
	Chungcheongnam-do	15	6.6
	Sejong	10	4.4
	Gyeongsangbuk-do	10	4.4
	Jeollanam-do	8	3.5
	Chungcheongbuk-do	6	2.6
	Daegu	6	2.6
	Ulsan	4	1.8
	Jeollabuk-do	3	1.3
	Gangwon-do	1	0.4

**Table 2 ijerph-19-05858-t002:** Job stress and job satisfaction according to general characteristics.

Variable			Job Stress				Job Satisfaction		
		*n*	Average	Standard Deviation	*p*-Value	*n*	Average	Standard Deviation	*p*-Value
Sex	Man	168	55.87	8.655	0.671	168	64.83	64.02	0.671
	Woman	59	56.44	9.475		59	64.02	11.57	
Age	Under 30	66	52.98	9.405	0.011	66	66.29	10.276	0.493
	30~39	71	57.35	8.647		71	62.96	13.52	
	40~49	69	57.01	8.522		69	64.72	12.377	
	Over 50	21	57.57	6.882		21	64.62	16.033	
Education	high school	4	54.00	1.414	0.848	4	67.00	4.690	0.746
	College	39	57.00	9.884		39	66.28	12.595	
	University	170	55.90	8.991		170	64.09	12.612	
	Graduate school	14	55.29	4.697		14	65.71	13.669	
Marriage	Single	121	54.44	8.826	0.006	121	65.02	11.328	0.023
	Married	97	58.16	8.866		97	63.16	13.801	
	Divorce/Bereavement/Separate	9	54.11	2.619		9	74.89	9.610	

**Table 3 ijerph-19-05858-t003:** Job stress and job satisfaction according to job-related characteristics.

Variable			Job Stress				Job Satisfaction		
		*n*	Average	Standard Deviation	*p*-Value	*n*	Average	Standard Deviation	*p*-Value
Qualification	Health manager	72	57.14	10.065	0.431	72	63.18	12.205	0.503
	Safety manager	154	55.50	8.246		154	65.29	12.732	
	Etc.	1	55.00			1	65.00		
Job title	Manager	37	59.84	7.006	0.010	37	63.57	12.620	0.819
	Middle manager	98	55.84	7.957		98	65.09	13.284	
	Worker	92	54.67	10.007		92	64.53	11.806	
Construction career	Less than 5 years	102	53.82	9.308	0.003	102	66.01	11.118	0.327
	5~10 years	42	57.60	8.082		42	64.10	14.262	
	10~20 years	58	56.95	8.363		58	64.09	12.848	
	More than 20 years	25	60.16	7.186		25	61.04	14.170	
Present career	Less than 5 years	149	54.95	9.596	0.015	149	65.63	12.511	0.015
	5~10 years	42	56.33	6.835		42	64.79	11.858	
	10~20 years	26	59.42	5.756		26	63.12	12.599	
	More than 20 years	10	61.80	7.786		10	52.70	10.884	
Department	Health and safety	226	56.02	8.877	0.909	226	64.62	12.578	0.976
	General affairs								
	Nursing								
	Infrastructure								
	Etc.	1				1			
Employment type	Full-time worker	82	57.74	7.817	0.007	82	62.85	13.408	0.020
	Temporary worker	142	55.28	9.219		142	65.27	11.852	
	Dispatch labor								
	Part-time worker								
	Etc.	3	43.67	2.887		3	64.62	12.550	
Average working hours per week	Less than 40 h	6	54.67	3.386	0.507	6	63.83	7.111	0.468
	41~48 h	52	54.77	8.961		52	66.37	12.803	
	49~56 h	145	56.21	8.953		145	63.67	12.902	
	More than 57 h	24	57.92	8.973		24	66.75	10.658	
Promotion opportunity	Possible	100	52.20	8.703	0.000	100	69.32	11.819	0.000
	Possible, but limited	95	58.72	7.548		95	61.96	10.801	
	Impossible	32	59.94	8.493		32	57.81	14.410	
Accident at work	Yes	28	58.86	7.132	0.070	28	63.79	14.433	0.709
	No	199	55.62	9.017		199	64.73	12.299	
Location city	Gyeonggi-do	37	56.46	8.620	0.266	37	64.08	11.196	0.008
	Seoul	92	55.93	8.183		92	64.72	11.882	
	Busan	16	56.75	9.406		16	67.25	15.102	
	Incheon	1	51.00			1	78.00		
	Chungcheongnam-do	15	57.07	8.614		15	60.13	12.501	
	Sejong	6	60.33	7.815		6	60.67	7.174	
	Gyeongsangbuk-do	10	51.20	7.285		10	73.70	7.587	
	Jeollanam-do	19	57.47	11.630		19	67.21	15.911	
	Chungcheongbuk-do	4	47.25	7.455		4	72.75	13.124	
	Daegu	6	62.00	9.879		6	51.00	11.349	
	Ulsan	10	58.60	8.383		10	55.30	13.573	
	Jeollabuk-do	8	49.88	9.877		8	68.38	5.208	
	Gangwon-do	3	51.33	2.082		3	70.67	9.238	

**Table 4 ijerph-19-05858-t004:** Effect of job satisfaction on job stress after controlling demographic characteristics.

Variable		Model 1				Model 2			
		B	SE	*β*	*t*(*p*)	B	SE	*β*	*t*(*p*)
(Constant)		42.122	11.607		3.629 *** (0.000)	84.950	9.140		9.294 *** (0.000)
Sex	Woman	1.641	1.841	0.081	0.891 (0.374)	2.694	1.356	0.134	1.987 * (0.048)
Age	Under 30	5.516	3.221	0.283	1.712 (0.088)	0.567	2.398	0.029	0.236 (0.813)
	30~39	6.954	2.967	0.365	2.344 (0.088)	2.476	2.207	0.130	1.122 (0.263)
	40~49	4.370	2.662	0.227	1.642 (0.102)	2.267	1.963	0.118	1.155 (0.250)
Education	College	1.437	4.793	0.061	0.300 (0.765)	4.750	3.532	0.203	1.345 (0.180)
	University	0.955	4.543	0.047	0.210 (0.834)	2.835	3.342	0.139	0.848 (0.397)
	Graduate school	0.330	5.046	0.009	0.065 (0.948)	0.584	3.709	0.016	0.157 (0.875)
Marriage	Single	1.549	3.453	0.087	0.449 (0.654)	−5.562	2.596	−0.314	−2.143 * (0.033)
	Married	4.848	3.426	0.271	1.415 (0.159)	−3.680	2.601	−0.206	−1.415 (0.159)
Qualification	Health manager	1.876	1.733	0.099	1.082 (0.280)	1.208	1.275	0.064	0.947 (0.345)
Job title	Middle manager	−0.848	2.195	−0.048	−0.386 (0.700)	−2.964	1.621	−0.165	−1.816 (0.071)
	Worker	−0.999	2.461	−0.055	−0.406 (0.685)	−3.733	1.821	−0.207	−2.050 * (0.042)
Construction career	Less than 5 years	−2.699	3.148	−0.152	−0.857 (0.392)	−1.947	2.314	−0.110	−0.841 (0.401)
	5~10 years	0.298	3.302	0.013	0.090 (0.928)	0.049	2.427	0.002	0.020 (0.984)
	10~20 years	−2.738	2.828	−0.135	−0.968 (0.334)	−2.821	2.079	−0.139	−1.357 (0.176)
Present career	Less than 5 years	−4.931	3.456	−0.265	−1.427 (0.155)	3.293	2.617	0.177	1.258 (0.210)
	5~10 years	−4.820	3.638	−0.212	−1.325 (0.187)	2.897	2.739	0.127	1.058 (0.291)
	10~20 years	−1.221	3.530	−0.044	−0.346 (0.730)	3.345	2.618	0.121	1.278 (0.203)
Department	Health and safety	5.768	8.642	0.043	0.667 (0.505)	3.922	6.354	0.029	0.617 (0.538)
Employment type	Full-time worker	8.814	5.056	0.479	1.743 (0.083)	−0.616	3.785	−0.034	−0.163 (0.871)
	Temporary worker	7.009	4.990	0.384	1.405 (0.162)	−1.350	3.723	−0.074	−0.362 (0.717)
Average working hours per week	41~48 h	2.613	3.612	0.124	0.723 (0.470)	1.334	2.657	0.063	0.502 (0.616)
	49~56 h	3.090	3.453	0.168	0.895 (0.372)	1.424	2.541	0.077	0.560 (0.576)
	More than 57 h	2.850	3.838	0.099	0.743 (0.459)	3.409	2.821	0.119	1.208 (0.228)
Promotion opportunity	Possible	−9.060	2.030	−0.509	−4.463 *** (0.000)	−1.372	1.604	−0.077	−0.855 (0.393)
	Possible, but limited	−3.350	1.924	−0.187	−1.741 (0.083)	−0.239	1.434	−0.013	−0.167 (0.868)
Accident at work	No	−0.010	1.885	0.000	−0.005 (0.996)	−0.982	1.387	−0.037	−0.708 (0.480)
Total score of job satisfaction						−0.490	0.038	−0.694	−13.053 *** (0.000)
*F*(*p*)		3.065 **				11.557 **			
*R* ^2^		0.294				0.620			
adj. *R*^2^		0.198				0.567			

* *p* < 0.05, ** *p* < 0.01, *** *p* < 0.001. Reference group: Gender*male, age*50 years, education level*high school*graduate, marriage status*divorce/divorce/separate/position*safety manager, position*manager, construction years*20 years, current position*more than 20 years, department*other, employment type*other, weekly working hours*40 h*no promotion opportunity.

**Table 5 ijerph-19-05858-t005:** Effect of job stress on job satisfaction after controlling demographic characteristics.

Variable		Model 1				Model 2			
		B	SE	*β*	*t*(*p*)	B	SE	*β*	*t*(*p*)
(Constant)		87.465	16.121		5.425 *** (0.000)	127.253	12.235		10.401 *** (0.000)
Sex	Woman	2.152	2.558	0.075	0.841 (0.401)	3.702	1.884	0.130	1.965 (0.051)
Age	Under 30	−10.107	4.474	−0.367	−2.259 * (0.025)	−4.897	3.313	−0.178	−1.478 (0.141)
	30~39	−9.144	4.121	−0.339	−2.219 * (0.028)	−2.576	3.070	−0.095	−0.839 (0.402)
	40~49	−4.295	3.697	−0.158	−1.162 (0.247)	−0.167	2.736	−0.006	−0.061 (0.951)
Education	College	6.767	6.658	0.204	1.016 (0.311)	8.124	4.894	0.245	1.660 (0.099)
	University	3.838	6.310	0.133	0.608 (0.544)	4.741	4.638	0.164	1.022 (0.308)
	Graduate school	0.519	7.009	0.010	0.074 (0.941)	0.831	5.152	0.016	0.161 (0.872)
Marriage	Single	−14.522	4.796	−0.579	−3.028 ** (0.003)	−13.059	3.527	−0.520	−3.703 *** (0.000)
	Married	−17.415	4.758	−0.688	−3.660 *** (0.000)	−12.836	3.515	−0.507	−3.652 *** (0.000)
Qualification	Health manager	−1.364	2.407	−0.051	−0.567 (0.571)	0.407	1.774	0.015	0.230 (0.819)
Job title	Middle manager	−4.283	3.049	−0.169	−1.405 (0.162)	−5.084	2.242	−0.201	−2.268 (0.024)
	Worker	−5.583	3.418	−0.219	−1.634 (0.104)	−6.527	2.513	−0.256	−2.597 ** (0.010)
Construction career	Less than 5 years	1.536	4.372	0.061	0.351 (0.726)	−1.013	3.219	−0.040	−0.315 (0.753)
	5~10 years	−0.507	4.587	−0.016	−0.111 (0.912)	−0.226	3.371	−0.007	−0.067 (0.947)
	10~20 years	−0.171	3.928	−0.006	−0.044 (0.965)	−2.757	2.894	−0.096	−0.953 (0.342)
Present career	Less than 5 years	16.796	4.800	0.637	3.499 *** (0.001)	12.138	3.546	0.460	3.423 *** (0.001)
	5~10 years	15.759	5.053	0.489	3.119 ** (0.002)	11.207	3.730	0.348	3.004 ** (0.003)
	10~20 years	9.325	4.903	0.237	1.902 (0.059)	8.172	3.604	0.208	2.267 * (0.024)
Department	Health and safety	−3.771	12.004	−0.020	−0.314 (0.754)	1.677	8.832	0.009	0.190 (0.850)
Employment type	Full-time worker	−19.260	7.022	−0.739	−2.743 ** (0.007)	−10.934	5.200	−0.419	−2.103 * (0.037)
	Temporary worker	−17.071	6.931	−0.660	−2.463 * (0.015)	−10.450	5.119	−0.404	−2.041 * (0.043)
Average working hours per week	41~48 h	−2.611	5.017	−0.088	−0.520 (0.603)	−0.143	3.692	−0.005	−0.039 (0.969)
	49~56 h	−3.403	4.796	−0.131	−0.709 (0.479)	−0.484	3.532	−0.019	−0.137 (0.891)
	More than 57 h	1.141	5.330	0.028	0.214 (0.831)	3.833	3.923	0.094	0.977 (0.330)
Promotion opportunity	Possible	15.699	2.820	0.622	5.568 *** (0.000)	7.142	2.174	0.283	3.286 *** (0.001)
	Possible, but limited	6.353	2.672	0.250	2.378 * (0.018)	3.188	1.979	0.126	1.611 (0.109)
Accident at work	No	−1.985	2.618	−0.052	−0.758 (0.449)	−1.994	1.924	0.052	−1.037 (0.301)
Total score of job stress						−0.945	0.072	−0.667	−13.053 *** (0.000)
*F(p)*		3.491 **				12.317 **			
*R* ^2^		0.321				0.635			
adj *R*^2^		0.229				0.584			

* *p* < 0.05, ** *p* < 0.01, *** *p* < 0.001. Reference group: Gender*male, age*50 years, education level*high school graduate*marriage status*divorce/divorce/separate/position*safety manager, position*manager, construction years*20 years, current position*more than 20 years, department*other, employment type*other, weekly working hours*40 h*no promotion opportunity*.

## Data Availability

Some or all data, models, or code generated or used during the study are proprietary or confidential in nature and may only be provided with restrictions. This study is based on a survey and contains human data, so data was secured through IRB approval.

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
