# Peer review of "Job Satisfaction Level of Safety and Health Manager in Construction Industry: Pandemic Period"

_ijerph, 2022, doi:10.3390/ijerph19105858_

Round 1

Reviewer 1 Report

The article concerns the assessment of job satisfaction of a group of safety and health protection managers in the construction industry. The issues raised by the authors of the publication include the assessment of the work of this professional group during the COVID-19 pandemic, i.e. it is a current topic and important from the point of view of, among others, construction management, or the assessment of engineers' working conditions.

Below are general comments about the article:

  1. According to the journal's requirements, the abstract should contain max. approx. 200 words, therefore it is necessary to shorten the abstract prepared for this article. Limit yourself to presenting the purpose of the publication, a short description of the research methodology and the most important conclusions from the analysis.
  2. The data presented in individual tables could be more legible, I mean the division of the table into individual rows, for example distinguishing data for each of the classifications separately (Table 1), the same applies to each of the following tables. Modifying the presentation of these data would facilitate the assessment of the issues studied.
  3. Maybe some survey data would be worth presenting not only in tabular form, but also in the form of graphs?
  4. It should be explained how the P-value criterion was assessed (significant, insignificant factors), it has not been discussed in every chapter. Maybe it would be good to explain in the beginning (in chapter 3) what does p <0.05, P <0.002, p <0.001 mean. The notation itself, for example "p <.05", is better shown as "p <0.05".
  5. In the final conclusions, it would be good to suggest, for example, how the job satisfaction of the surveyed employees can be influenced, which would facilitate and improve the standard of their work.
  6. The formatting of the references should be corrected - the formatting of the font itself (e.g. italics in appropriate places), lack of complete information for some publications (e.g. publication no. 21, 29).

Author Response

1. According to the journal's requirements, the abstract should contain max. approx. 200 words, therefore it is necessary to shorten the abstract prepared for this article. Limit yourself to presenting the purpose of the publication, a short description of the research methodology and the most important conclusions from the analysis.

☞ In accordance with the comments pointed out by you, abstract was rewritten in 199 words. Please refer to page 1.

2. The data presented in individual tables could be more legible, I mean the division of the table into individual rows, for example distinguishing data for each of the classifications separately (Table 1), the same applies to each of the following tables. Modifying the presentation of these data would facilitate the assessment of the issues studied.

☞ We revised all the tables according to your suggestion. We drew lines for each classification in the table and separated them to improve the visibility.

3. Maybe some survey data would be worth presenting not only in tabular form, but also in the form of graphs?

☞ I thought about expressing the data in a graph. However, since there are a lot of survey questions and we don't represent a continuous trend, we decided that it would be better to tabulate them.

4. It should be explained how the P-value criterion was assessed (significant, insignificant factors), it has not been discussed in every chapter. Maybe it would be good to explain in the beginning (in chapter 3) what does p <0.05, P <0.002, p <0.001 mean. The notation itself, for example "p <.05", is better shown as "p <0.05".

☞ We added the significance level to the 2.3 data analysis section(Line 135 to 137) in accordance with your opinion. And all the notations were also modified.

5. In the final conclusions, it would be good to suggest, for example, how the job satisfaction of the surveyed employees can be influenced, which would facilitate and improve the standard of their work.

☞ In the conclusion section, we added the work level to improve by strengthening the job satisfaction survey in jobs with high intensity of work such as construction(Line 344 to 347).

6. The formatting of the references should be corrected - the formatting of the font itself (e.g. italics in appropriate places), lack of complete information for some publications (e.g. publication no. 27, 35).

☞ All the reference formats have been rewritten according to the format of "IJERPH". Publication no. 27 and 35 have been changed into full information. Thank you for your valuable opinion. Thank you very much for your valuable comments.

Reviewer 2 Report

The Authors deal with an interesting topic and the present paper describes very important issue. My main issue is that the structure of the paper is very poor and does not represent a scientific research paper. Unfortunately, there are several insufficiencies that need to be improved. 

1. the title is interesting, although it may be a bit ambitious and too long.

2. the abstract is too long. According to the "Instructions for Authors": The abstract should be a total of about 200 words maximum. The abstract should be a single paragraph and should follow the style of structured abstracts, but without headings: 1) Background: Place the question addressed in a broad context and highlight the purpose of the study; 2) Methods: Describe briefly the main methods or treatments applied. Include any relevant preregistration numbers, and species and strains of any animals used. 3) Results: Summarize the article's main findings; and 4) Conclusion: Indicate the main conclusions or interpretations. The abstract should be an objective representation of the article: it must not contain results which are not presented and substantiated in the main text and should not exaggerate the main conclusions.

3. The introduction (Section 1) describe the research background very poor. The articles cited are not known in the international literature, they have no citations. The paper does not present the full background of the issue under study. There are many more papers on this topic and the literature review should be improved. 

4. The Section 2 (Method) should be dedicated to describing this methodology and what you did in your paper. The methodology should be described and be solid enough such that any other person using the same procedure will could repeat the research. Now, it is impossible. Now this section contains only short description of the method and nothing else. 

5. I do not see connection between the aim of the research and the pandemic situation. I think that this paper is not the scientific paper - it is only case study and description local situation in Korea.

6. in this paper I do not see very important section: Discussion. Unfortunately, this paper does not point out the shortcomings of past research to show the value of this research. Add a strengths and weaknesses section and limitations section of this research to the new section: Discussion. The discussion should refer to other studies, indicate the shortcomings of the research. The research has some limitations. The manuscript should highlight some of these limitations.

7. in the conclusion, the manuscript should discuss the practical applications and implications of the research.

8. possible areas for future research should be discussed.

9. I believe that this is not a scientific article, just the description of innovation.

Overall, at the moment the manuscript does not reach the desired level for publishing. I strongly urge the Authors to reconsider the above-mentioned comments, rewrite the paper accordingly, and resubmit

Author Response

1. the title is interesting, although it may be a bit ambitious and too long.

☞ Based on your comment, the title has been shortened to: "Job satisfaction level of safety and health manager in construction industry: Pandemic period"

2. the abstract is too long. According to the "Instructions for Authors": The abstract should be a total of about 200 words maximum. The abstract should be a single paragraph and should follow the style of structured abstracts, but without headings: 1) Background: Place the question addressed in a broad context and highlight the purpose of the study; 2) Methods: Describe briefly the main methods or treatments applied. Include any relevant preregistration numbers, and species and strains of any animals used. 3) Results: Summarize the article's main findings; and 4) Conclusion: Indicate the main conclusions or interpretations. The abstract should be an objective representation of the article: it must not contain results which are not presented and substantiated in the main text and should not exaggerate the main conclusions.

☞ According to your opinion, the abstract was rewritten with 199 words. Thank you for explaining how to write in detail. Please refer to page 1

3. The introduction (Section 1) describe the research background very poor. The articles cited are not known in the international literature, they have no citations. The paper does not present the full background of the issue under study. There are many more papers on this topic and the literature review should be improved.

☞ We added the contents of the research background to Introduction by reflecting your opinion (Line 66 to 72).

4. The Section 2 (Method) should be dedicated to describing this methodology and what you did in your paper. The methodology should be described and be solid enough such that any other person using the same procedure will could repeat the research. Now, it is impossible. Now this section contains only short description of the method and nothing else.

☞ Thank you for your opinion. In order to create an additional methodology, we considered adding the contents of the questionnaire as a picture, but it seems to force the volume of the paper to increase. The questionnaire applied to this study is almost the same questionnaire that was previously commercialized. So we think it is right to replace it with a reference. If it's still necessary, I'll submit it as a separate appendix. The contents of the distribution, collection, and analysis of the questionnaire would be sufficient from 2.1 to 2.3 of "The Section 2 Method".

5. I do not see connection between the aim of the research and the pandemic situation. I think that this paper is not the scientific paper - it is only case study and description local situation in Korea.

☞ We agree to your opinion. However, it is natural for the case study to reflect the local situation at that time. We think data accumulation through case studies in a special situation called as “Pendemic” is an essential prior study for future countermeasures and comparative research with other countries.

6. in this paper I do not see very important section: Discussion. Unfortunately, this paper does not point out the shortcomings of past research to show the value of this research. Add a strengths and weaknesses section and limitations section of this research to the new section: Discussion. The discussion should refer to other studies, indicate the shortcomings of the research. The research has some limitations. The manuscript should highlight some of these limitations.

☞ We agree with you on the creation of limitations. Therefore, the limitations of the study were additionally written at the last part (Line 324 to 331) of Discussion. However, it is inappropriate to point out the shortcomings of previous studies because there are no studies on job satisfaction of construction workers in the past that have considered modern and contemporary pandemics.

7. in the conclusion, the manuscript should discuss the practical applications and implications of the research.

☞ According to your suggestion, we added it to the conclusion (Line 344 to 347).

8. possible areas for future research should be discussed.

☞ It was added to "Discussion" (Line 314 to 316).

9. I believe that this is not a scientific article, just the description of innovation.

☞ In the construction industry, the treatment of workers has been very poor since long ago, and the evaluation of their job satisfaction has been negative and extremely rare. In addition, in South Korea, safety managers and health managers are divided and the two jobs are clearly different within the construction site. In addition, since Pendemic is a very special situation, this study is judged to be valuable only with the above three conditions. If similar research is conducted later, it should be used as basic data. Thank you very much for your valuable feedback.

Reviewer 3 Report

The paper discussed the job satisfaction of H&S managers is interesting and aligns with the aim of IJERPH. There are some concerns addressed below. 

  1. There is a need to proofread the paper carefully before publication.
  2. Need to clear the purpose of the work. What is your primary focus? The workers or the H&S managers?
  3. From line 92 to line 99. You are trying to investigate job stress and job satisfaction over the workers but you selecting the H&S managers because you are saying they work both in the field and office. This is not a proper reason. As the H&S managers’ samples may have differences in multidimensions than the workers or the office people, including work pressure, especially in the Pandemic situation.
  4. Survey questions need to be presented in the paper.
  5. The description of survey items is not scientific enough (Line 120 – Line 131).
  6. Line 276 to 270, why salary is not included in the current work? Please give reasons.
  7. In the beginning of the paper, the authors declare that the aim of the work is to determine the significant factors of job satisfaction and job stress, and this is for finding a way to reduce the turnover rate. This should be further discussed in Section 3.5. However, in the current issue, this is weakly addressed. Need more discussion e.g. how to improve employment stability, how to improve the trust of the organization, or how to deliver promotion opportunities.
  8. Many managers in the workplace are dealing with health and safety together, how did you analyze them? Do all the managers in the work do safety and health separately?
  9. Incomplete reference information, for example, No. 12 and 21.
  10. Line 325 to 326, the sentence needs to improve.
  11. I think maybe there is a need to change the title of the work. As the current work is not aiming to address the level of satisfaction of H&S managers but the factor that affects satisfaction and the stress most.
  12. It is weakly addressed why the parameters contribute to job satisfaction and job stress in the pandemic situation (COVID-19). For example, the finding of Model 1 and Model 2, e.g. promotion opportunities, as they are also important in a normal work situation.

Author Response

1. There is a need to proofread the paper carefully before publication.

☞ According to your suggestion, we revised many parts of the paper along with the opinions of other reviewers.

2. Need to clear the purpose of the work. What is your primary focus? The workers or the H&S managers?

☞ The subjects of this study are health managers and safety managers. Except for one other person, 226 people belong to them.

3. From line 83 to line 95. You are trying to investigate job stress and job satisfaction over the workers but you selecting the H&S managers because you are saying they work both in the field and office. This is not a proper reason. As the H&S managers’ samples may have differences in multidimensions than the workers or the office people, including work pressure, especially in the Pandemic situation.

☞ The purpose of this study is to understand the overall job satisfaction of construction workers. In Korea's construction industry, office workers are almost 100 percent Korean, and many outdoor workers consist of foreign workers. In Korea, safety managers and health managers are separated, and their duties are very different. Therefore, the research was conducted based on the judgment of health managers and safety managers for jobs that are easy to collect survey data and can replace both office and outdoor workers. I agreed with Reviewer3 and added it to Introduction (Line 87 to 92).

4. Survey questions need to be presented in the paper.

☞ The volume of the paper is too much to include all the questions in the survey. And the questionnaire used in this study is a commercialized questionnaire for evaluating the job satisfaction of satisfaction. So I decided that it was unnecessary to write all the questions separately. However, if it is necessary to put it in, I will consider submitting it as an appendix.

5. The description of survey items is not scientific enough (Line 120 – Line 131).

☞ The questions in the survey were replaced with references. The contents of the distribution, collection, and analysis of the questionnaire were judged to be sufficient from 2.1 to 2.3 of "The Section 2 Method".

6. Line 276 to 270, why salary is not included in the current work? Please give reasons.

☞ In South Korea, salary in the workplace is very personal information. The Korean Construction and Health Association, which distributed and collected the survey, wanted to delete the salary to protect the privacy of workers. We added the above to Discussion because we think your inquiry is important (Line 276 to 279).

7. In the beginning of the paper, the authors declare that the aim of the work is to determine the significant factors of job satisfaction and job stress, and this is for finding a way to reduce the turnover rate. This should be further discussed in Section 3.5. However, in the current issue, this is weakly addressed. Need more discussion e.g. how to improve employment stability, how to improve the trust of the organization, or how to deliver promotion opportunities.

☞ In accordance with your opinion, we added the limitations of this study as well as the need for further research to “Discussion” (Line 314 to 316).

8. Many managers in the workplace are dealing with health and safety together, how did you analyze them? Do all the managers in the work do safety and health separately?

☞ In South Korea, safety managers and health managers are hired separately and their duties are also different.

9. Incomplete reference information, for example, No. 17 and 27.

☞ Reference publication No. 17 and 27 have been changed to complete information.

10. Line 337 to 339, the sentence needs to improve.

☞ Thank you for your feedback. It has been modified.

11. I think maybe there is a need to change the title of the work. As the current work is not aiming to address the level of satisfaction of H&S managers but the factor that affects satisfaction and the stress most.

☞ Other reviewers also suggested to revise the title concisely, so the title has been modified as follows: "Job satisfaction of the safety and health manager in the construction industry: Pandemic period."

12. It is weakly addressed why the parameters contribute to job satisfaction and job stress in the pandemic situation (COVID-19). For example, the finding of Model 1 and Model 2, e.g. promotion opportunities, as they are also important in a normal work situation.

☞ In this study, it was not possible to understand how to contribute to job satisfaction by simply asking about the presence or absence of promotion opportunities. In accordance with your opinion, therefore, we wrote the need for further research on promotion opportunities in Discussion (Line 281 to 284).

Thank you very much for your valuable comments.

Round 2

Reviewer 2 Report

I would like to thank the Authors for respond and changes , but I still think that the structure of the paper is very poor and does not represent a scientific research paper. Unfortunately, there are still several insufficiencies that need to be improved. 

3. The introduction (Section 1) describe the research background very poor. The articles cited are not known in the international literature, they have no citations. The paper does not present the full background of the issue under study. There are many more papers on this topic and the literature review should be improved. The added literature are not enough.

4. The Section 2 (Method) should be dedicated to describing this methodology and what you did in your paper. The methodology should be described and be solid enough such that any other person using the same procedure will could repeat the research. Now, still it is impossible. The Authors added only one new sentence... the purpose of describing the methodology has not yet been achieved

5. I do not see connection between the aim of the research and the pandemic situation. I think that this paper is not the scientific paper - it is only case study and description local situation in Korea. The Authors agreed with my opinion, but the paper is still "article" not "case study".

Overall, at the moment the manuscript does not reach the desired level for publishing. I strongly urge the Authors to reconsider the above-mentioned comments, rewrite the paper accordingly, and resubmit

Author Response

  1. The introduction (Section 1) describe the research background very poor. The articles cited are not known in the international literature, they have no citations. The paper does not present the full background of the issue under study. There are many more papers on this topic and the literature review should be improved.The added literature are not enough.

☞ According to your comment, we added more literature to increase the background of the study (Line 48 to 56)

  1. The Section 2 (Method) should be dedicated to describing this methodology and what you did in your paper. The methodology should be described and be solid enough such that any other person using the same procedure will could repeat the research. Now, still it is impossible. The Authors added only one new sentence... the purpose of describing the methodology has not yet been achieved

☞ According to your opinion, the contents of recruitment of research subjects, anonymity guarantee method, research consent, and elimination criteria were written in as much detail as possible in the methodology(Line 111 to 122). And we attached the questionnaire as an additional file.

  1. I do not see connection between the aim of the research and the pandemic situation. I think that this paper is not the scientific paper - it is only case study and description local situation in Korea. The Authors agreed with my opinion, but the paper is still "article" not "case study".

☞ We revised the research goal and rewritten it in Introduction(Line 97 to 105). Interpretations of the results obtained from this study and a solution for each factor affecting job satisfaction were prepared in “Discussion”(Line 292 to 325). In addition, we discussed several ways to improve employment stability and organizational trust and deliver promotion opportunities and added it to “Discussion”. Additionally we will ask the editorial board to designate the type of our paper as not “article” but “case study(or report)” in accordance with your opinion. Thank you for your valuable review.

Reviewer 3 Report

See the attached below. 

Author Response

  1. Please, include questionnaire questions into the appendix, this would be good for other academics to adopt your method in the future.

☞ According to your opinion, we translated the questionnaire used in this study into English and attached it to additional materials (appendix).

  1. You did not response the concern well – “In the beginning of the paper, the authors declare that the aim of the work is to determine the significant factors of job satisfaction and job stress, and this is for finding a way to reduce the turnover rate. This should be further discussed in Section 3.5. However, in the current issue, this is weakly addressed. Need more discussion e.g. how to improve employment stability, how to improve the trust of the organization, or how to deliver promotion opportunities.

” I think if you address the above question well then this could be your potential contribution of the work, and you were saying so in the previous version of the paper.

But now you declared this is your limitation, then I am start questioning the contribution of the work. Without this the work looks like a report, a paper is looking for some more critical findings and solutions.

As the findings of the work now is that “The highest among unmarried, divorced, bereavement, and separated; the higher the rank, the higher the job stress, and the higher the job stress when the construction industry worked for a long time.”

How would this help the industry? Would these factors affect satisfaction different from a normal situation (non-pandemic situation)? Actually, these factors would always affect a person’s motion at any time. Would these be significant in a pandemic situation? You haven’t discussed this. Also, these factors are unlikely be managed at workplace, then how the industry people should response based on the findings? I think you should at least discuss this. A comparison of the findings (pandemic period) with previous work (normal situation) may be helpful.

☞ We revised the research goal and rewritten it in Introduction(Line 97 to 105). Interpretations of the results obtained from this study and a solution for each factor affecting job satisfaction were prepared in “Discussion”(Line 292 to 325). In addition, we discussed several ways to improve employment stability and organizational trust and deliver promotion opportunities and added it to “Discussion”. Thank you for your valuable review.
